# An enhanced multilevel secure data dissemination approximate solution for future networks

**Mohammad Mahmood Otoom**[1]*, **Mahdi Jemmali**[3,4,5], **Akram Y. Sarhan**[6], **Imen Achour**[1], **Ibrahim Alsaduni**[2], **Mohamed Nazih Omri**[5]

1 Department of Computer Science and Information, College of Science at Zulfi, Majmaah University, Al-Majmaah, Saudi Arabia, 2 Department of Electrical Engineering, College of Engineering, Majmaah University, Al-Majmaah, Saudi Arabia, 3 College of Computing and Informatics, University of Sharjah, Sharjah, United Arab Emirates, 4 Higher Institute of Computer Science and Mathematics, University of Monastir, Monastir, Tunisia, 5 Mars Laboratory, University of Sousse, Sousse, Tunisia, 6 Department of Information Technology, College of Computing and Information Technology at Khulis, University of Jeddah, Jeddah, Saudi Arabia

* m.otoom@mu.edu.sa

**Data Availability Statement:** All relevant data are within the paper.

**Funding:** The funders had no role in study design, data collection and analysis, decision to publish, or

## Abstract

Sensitive data, such as financial, personal, or classified governmental information, must be protected throughout its cycle. This paper studies the problem of safeguarding transmitted data based on data categorization techniques. This research aims to use a novel routine as a new meta-heuristic to enhance a novel data categorization based-traffic classification technique where private data is classified into multiple confidential levels. As a result, two packets belonging to the same confidentiality level cannot be transmitted through two routers simultaneously, ensuring a high data protection level. Such a problem is determined by a non-deterministic polynomial-time hardness (NP-hard) problem; therefore, a scheduling algorithm is applied to minimize the total transmission time over the two considered routers. To measure the proposed scheme's performance, two types of distribution, uniform and binomial distributions used to generate packets transmission time datasets. The experimental result shows that the most efficient algorithm is the Best-Random Algorithm ($\widetilde{BR}$), recording 0.028 s with an average gap of less than 0.001 in 95.1% of cases compared to all proposed algorithms. In addition, $\widetilde{BR}$ is compared to the best-proposed algorithm in the literature which is the Modified decreasing Estimated-Transmission Time algorithm ($MDETA$). The results show that $\widetilde{BR}$ is the best one in 100% of cases where $MDETA$ reaches the best results in only 48%.

## Introduction

The technological advancement in this era has made collaboration among different governmental and none governmental organizations to be a necessity. Such collaboration requires secure and private data dissemination infrastructure. Multilevel Data security has been proposed to increase data security and control the disclosing of data in a particular environment. However, there are not enough efforts in secure data classification research. On the one hand,

preparation of the manuscript. The authors extend their appreciation to the deputyship for Research & Innovation, Ministry of Education in Saudi Arabia for funding this research work through the project number (IFP-2022-33).

**Competing interests:** NO authors have competing interests.

most existing solutions that rely on traditional security mechanisms are less effective since common security weaknesses are inherited from the previous Open System Interface (OSI) model. On the other hand, there is a need for secure-by-design efforts to eliminate the gap between previous weaknesses and current advancements to enable secure future networks. Hence, this research effort improves the previous research that relies on a novel secure-by-design network model that uses two routers for secure classified data dissemination. The improvement focused on minimizing the total transmission time over the two considered routers.

Emerging technology has led to the Fifth Industrial Revolution. It is evolving rapidly due to numerous advancements in many fields, such as system intelligence, machine learning, IoT, robotics, blockchain, and so forth. Furthermore, these advancements cause the world to become more interconnected by integrating various new networking technologies, including smart devices with the Internet. In addition, the outbreak of an incident like COVID-19 has accelerated the efforts in technology research and development.

The number of devices connected to the Internet has been increasing. For instance, more than ten billion devices are connected today [1, 2]. The design of interconnected networks relies on the Open Systems Interface (OSI) model. Although many advantages are incurred from the OSI model, such as protocol integration, standardization, flexibility, and modularity, the complexity of security requirements makes the OSI model's current design less effective in dealing with numerous attacks. For example, active, passive, and advanced network attacks usually degrade the network performance causing the packet traffic to be uncontrolled. Such common attacks may include spoofing, denial of service (DoS), sinkholes, Sybil, traffic analysis, eavesdropping, byzantine, black hole, and location disclosure attacks [3]. These attacks could cause severe threats to safeguarding sensitive transmitted data.

The number of security incidents, vulnerabilities, cybersecurity threats, and challenges has increased with the introduction of Industry 5.0. In particular, IoT, digitalization, big data, and artificial intelligence are considered the primary causes of cybersecurity issues since they have notably increased internet connection use. As a result, attacks have rapidly expanded to target significant businesses and industries, including health organizations, to disrupt infrastructure and steal sensitive data [4]. These attacks may include email fraud and spam, network anomalies, malicious URLs, impersonation, and malware attacks [5, 6].

Some network attacks, like Sybil attacks and Eclipse attacks, target the security of routers by damaging their functionality. Some of these widespread attacks are as follows: Routing Table Poisoning attacks (RTP), Packet Mistreating Attacks (PMA), Hit and Run attacks (HAR), Advanced Persistent Threats (APT) (PA), and Denial of Service attacks (DoS) also fall under widespread attacks. For example, the Border Gateway Protocol (BGP) has a critical routing target on the Internet. It manages the routing of packets throughout the Internet via exchanging information among edge routing. However, it does not maintain security and is vulnerable to attacks [7, 8].

This paper proposes an intelligent private network solution that uses a security policy to provide a multilevel data dissemination control based on security category constraints. While on the one hand, the elaboration of the proposed solution relies on the scheduling algorithms presented in [9–11], which are planned later to be utilized and adapted to the studied problem. On the other hand, applying the security category constraints idea in this paper is considered an NP-hard problem, which means that no algorithm will always efficiently produce the exact correct answer on all inputs. Therefore, the scheduling algorithm is the best choice to get the best solution in the current work. The packets will be scheduled to be waiting in a queue or need to be transmitted. This paper aims to introduce a future secure computer network paradigm. The model includes a packet dissemination security policy to minimize or prevent data

breaches. Several real-life scenarios, case studies, and applications can benefit from the model, including the media industry, military organizations, and journalism. The proposed two-router network is an NP-hard problem [12] that uses a security policy for multilevel data security transmission. However, it has many benefits: (i) introduces a security-by-design future network idea that relies on a dissemination-based packet classification; (ii) applies various algorithmic techniques such as dispatching rules, local insertion search, randomization method, and lifting procedure to enhance the security and performance of the transmitted packets in the computer network area; (iii) presents an idea that can be implemented as a private network application to assist individual privacy protection in critical environments and circumstances.

The novelty of this research is its security by design paradigm that initially relies on two routers for future network security. For example, the solution supports journalists for multi-level secure and successful transmission of their data in the form of network packets. Furthermore, the proposed work introduces several algorithmic solutions to deal with an NP-Hard problem in acceptable efficient time and use it in the network security field. Furthermore, compared with previous works. The proposed approach continues to enhance the results compared with previous developed algorithms in [12–14] results as it being detailed in the result and discussion section. It uses new algorithmic techniques and procedures. On the other hand, the disadvantages of the proposed method are as follows: (i) the time complexity for such an NP-hard two-machine problem demands using more techniques to solve the problem using algorithms with $O(n^3)$ heuristically. (ii) Developing an exact solution requires using a lower bound in a branch-and-bound algorithm.

This paper is structured as follows. First, section two presents the related works. Next, section three presents the description of the studied problem. Then, section four describes the solution novel architecture through an illustrative diagram. Section five details the enhancement randomization routine, and section six describes the proposed algorithms and their different instructions. Section seven provides the experimental results and discusses the solution performance evaluation. Finally, section eight concludes the remarks and future works.

## Related works

Secure and private data outsourcing or exchanging have primarily been studied in the literature at the application layer. Several known techniques or traditional methods have been applied to protect such data. However, the innovation of the proposed approach in this paper is that it addresses the problem from a different point of view by providing a solution that can be applied at the network layer and introducing heuristic solutions reduced from a known NP-Hard problem to address issues in the field of network security.

The studied problem was addressed for the first time in [12], in which only dispatching rules techniques were used to construct algorithmic solutions. However, this paper uses several sophisticated algorithms for the problem. In addition, authors in [13] proposed several algorithms based on critical-level security and randomization to deal with the studied problem. In the same context, the authors in [14, 15] proposed novel heuristics to solve the studied problem approximately. The author in last work used several recognized and unknown algorithmic methods like iterative, randomization, and probabilistic methods. The achieved results that both proposed algorithms called $RGS_1$ and $RGS_2$ were better compared with previous proposed algorithms that deal with same studied problem.

The literature has investigated several threats to data transmission across all network layers in various distributed computing systems, applications, and technologies. The analysis of such threats has shown attacks in various emerging and intelligent communications and

technologies domains. For example, in [16], wireless mobile ad hoc network attacks are investigated. In [17], wireless sensor network attacks are discussed. Finally, the authors in [18] addressed attacks affecting SDN and cloud computing environments.

Several works related to the representation of network traffic have been developed [19, 20]. For example, the constraint of the window pass is proposed for the first time in [22] to prioritize highly confidential packets. In addition, the one router problem is studied under fixed time-slot interval in [21, 22] to experiment with different developed algorithms and thus prove the efficiency of the presented work. Finally, in [23], the authors introduced a scheduler component to solve the problem of parallel routers in the network.

Multilevel data security (MLDS) [24] has become an essential tool since the new emerging technology caused organizations to collaborate to securely share algorithms and data or jointly process the data to extract valuable knowledge. In MLDS, data is labeled and accessed according to their critical level. Hence, there is a need for a multilevel secure dissemination solution in the multi-domain environment [25]. Furthermore, such a solution is needed in a military-based climate since no sufficient research exists in this domain. There is no practicality in the current deployed access-control methods for implementing secure data dissemination for data streams [26, 27].

The algorithms developed in [28] can be extended and applied to the presented problem. The algorithms of the parallel machine problem [29–32] can be extended and reformulated to use the related algorithms for the studied problem. Moreover, the algorithms developed in this paper can be extended for the subject considered in [33, 34]

The issues of packet scheduling, timing, and routing have been well-studied by many. For example, several routing scheduling protocols have been proposed to deal with packet timing minimization and queuing issues since additional issues have added to the network routing problems due to innovative advancements in communications technologies. The common packet routing issues are energy consumption, optimization, latency, overhead, routing security, and privacy. Thus, many studies proposed to deal with issues presented in modern network technology [35–38]. The literature presented several algorithms regarding the scheduling problem in networks.

Several algorithms have been proposed to deal with network scheduling problems. However, algorithms presented in [12–15] dealt with the issue presented in this paper which is transmitting multiple levels of data based on a constraint. Such a problem is an NP-hard [12] because the minimization of total time for transmitting data through two routers in this paper is reduced from two parallel machines NP-Hard problem.

## Problem description

The description of the proposed problem is as follows. A set of files in a special network that each has different security characteristics must be transmitted through two routers. Suppose each set of files is classified according to a security level denoted by $Sl_i$ with $i = \{1, \ldots, n_{Sl}\}$ where $n_{Sl}$ means the number of security levels. The files are then split into different packets categorized according to their corresponding $Sl_i$. Let $Pcs$ be the set of all packets, and $n_{pcs}$ is the total packet number. $Pk_j$ denotes the packet with index $j$. The security level of the packet $Pk_j$ denotes by $Dlp_j$. Each packet $Pk_j$ has its own transmission time denoted as $Pt_j$. Based on packet $Pk_j$, the cumulative transmission time on the first router $R_1$ and the second router $R_2$ are denoted by $CT_1^j$ and $CT_2^j$, respectively. The set of packets transmitted through $R_1$ and $R_2$ are denoted by $Pcs_1$ and $Pcs_2$, respectively. Therefore, $Pcs = Pcs_1 \cup Pcs_2$ and $n_{pcs} = |Pcs_1| \cup |Pcs_2|$. After the transmission of packets, the total transmission time on router $R_1$ is denoted by $Tt_1$, and the one on $R_2$ is denoted by $Tt_2$. Thus, the maximum transmission time for the two routers

denotes by $Tt_{max}$ and given by $Tt_{max} = \max(Tt_1, Tt_2)$. The objective is to minimize $Tt_{max}$. Packets holding the same security levels cannot be transmitted simultaneously through the two routers. Therefore, a security constraint is being used to prevent the transmission of two packets belonging to the same security level simultaneously through two different routers. Such a problem has been proven to be NP-hard [14].

## Architecture

This section defines a novel architecture based on the proposed constraint of security levels. This architecture comprises six components, as Fig 1 explains. The "Data security level" and the "Scheduler" are the two main components. The administrator imposes and links each file to their security level. The decomposition of each file into packets will automatically give the authorization of the component "Data security level" to add the same security level of the same file for all packets belonging to this file.

## Enhancement randomization routine (*ER*2)

The section presents a randomization-routine enhancement called to improve the solution of the developed algorithm within a deployed scheduler. The routine relies on a probability

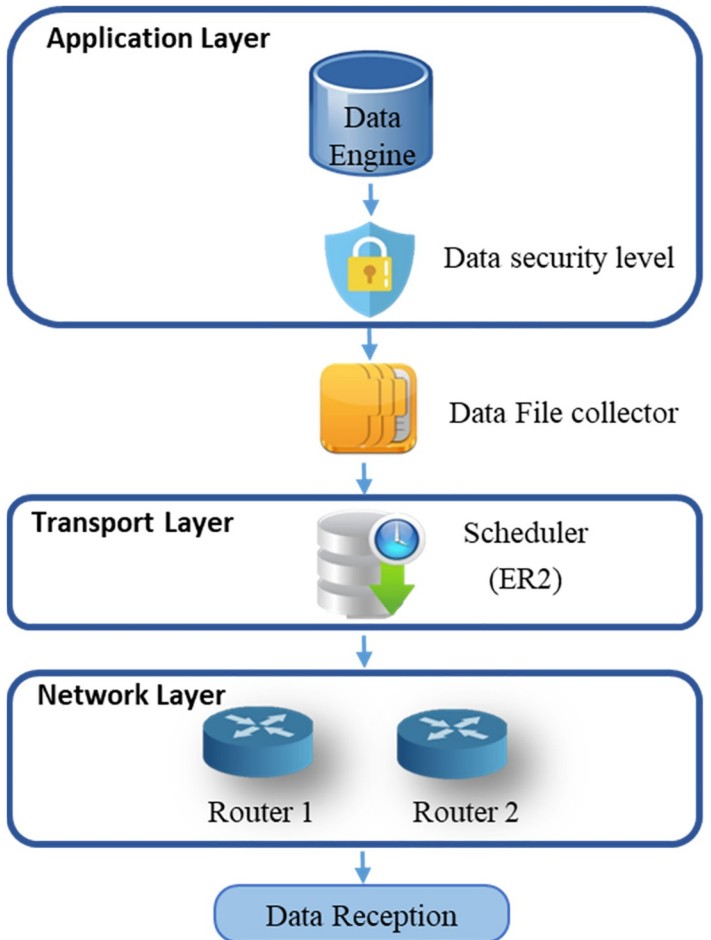

**Fig 1.**

**Table 1. Decreasing time algorithm sequence.**

| $j$ | 3 | 1 | 7 | 4 | 8 | 11 | 5 | 12 | 6 | 9 | 13 | 2 | 10 |
|---|---|---|---|---|---|---|---|---|---|---|---|---|---|
| $Sl_i$ | 3 | 1 | 4 | 4 | 3 | 2 | 4 | 4 | 4 | 2 | 1 | 2 | 2 |
| $Pt_j$ | 27 | 25 | 18 | 13 | 13 | 11 | 10 | 10 | 4 | 3 | 3 | 1 | 1 |

**Table 2. The sequence after applying $ER2(.)$.**

| $j$ | 3 | 1 | 7 | 4 | 8 | 11 | 5 | 12 | 6 | 9 | 13 | 2 | 10 |
|---|---|---|---|---|---|---|---|---|---|---|---|---|---|
| $r$ | 41 | 67 | 34 | 0 | 69 | 24 | 78 | 58 | 62 | 64 | 5 | 45 | |
| $j'$ | 3 | 1 | 7 | 8 | 4 | 5 | 11 | 12 | 6 | 9 | 2 | 13 | 10 |
| $Sl_i$ | 3 | 1 | 4 | 4 | 3 | 2 | 4 | 4 | 4 | 2 | 1 | 2 | 2 |
| $Pt_j$ | 27 | 25 | 18 | 13 | 13 | 10 | 11 | 10 | 4 | 3 | 1 | 3 | 1 |

method. The corresponding sequence for any schedule obtained by any heuristic $h()$ is determined. The sequence is stored in a list with two selection choices that select either the first or the second packet. This choice is based on the probability $\beta$. Indeed, to choose the first packet, the probability $\beta$ and the probability $1 - \beta$ is applied for the second packet. Hereafter, $\tilde{h}$ is denoted by the heuristic value obtained after calling the enhancement randomization routine $ER2(.)$ by inputting the sequence obtained by the heuristic $h$. Thus, $\tilde{h} = ER2(h)$. For each algorithm, the call of $ER2(.)$ will be executed 500 times, and the best solution will be picked.

**Example 1**. *Suppose that the number of packets is 13. Then, the sequence obtained after utilizing the decreasing time algorithm is presented in* Table 1.

*After calling ER2(.) described above, an obtained sequence is presented in* Table 2. *In this latter table, r is a number in [1, 100]. When this number exceeds 30, the first packet is chosen; otherwise, the second one will be chosen. In addition, j' is the packet index in the new sequence after enhancement.*

*Based on* Table 2, *the schedule of the proposed problem is presented in* Fig 2. *The latter figure shows that on the first router, the packets {4, 5, 12, 6} are scheduled, however on router 2 packet 7 is scheduled respecting the constraint of the data security level.*

## The proposed algorithms

The section presents seven proposed algorithms and their details. The enhancement of the algorithms relies on the above novel routine ($ER2$). The seven enhanced algorithms are called the random-decreasing time algorithm, the random-modified decreasing time algorithm, the random-search and insertion time algorithm, the random-critical security level algorithm, the random packet-classification first variant algorithm, the random packet-classification second variant algorithm, and the best-random algorithm.

### Random-decreasing time algorithm ($\widetilde{DT}$)

In this algorithm, packets are sorted based on their transmission time-decreasing order. Then, packets are scheduled one by one. Thus, the first packet is selected and scheduled on the router with the minimum values among $CT_1^j$ and $CT_2^j$ then the second packet is scheduled the same way, and the same applies to the remaining packets until all are scheduled. Finally, after the accomplishment of the scheduling, this algorithm denotes $DT$. The sequence obtained by $DT$ will be used as the initial solution for applying $ER2$. This algorithm denotes $\widetilde{DT}$.

## Random-modified decreasing time algorithm ($\widetilde{MD}$)

The packets are sorted for this algorithm, as detailed in the previous subsection. When a packet $Pk_j$ is selected, each interval time for all routers where there is no packet in transmission will be called "Idle Interval" and will be detected by this algorithm. The "Idle Interval" is denoted by $Ii$. A test of the load is applied by calculating $CT_1^j$ and $CT_2^j$. If $CT_1^j = CT_2^j$, then the selected router will be the one that has the shortest idle interval. The algorithm that returned the sequence of $MD$ is detailed in Algorithm 1. After finishing the scheduling, the obtained sequence will be used as the initial solution for applying ($ER2$). This algorithm denotes by $\widetilde{MD}$. Hereafter, DCR($L$) denotes the procedure that sorts a list $L$ given as input according to their transmission time-decreasing order.

**Algorithm 1** Modified Decreasing Time Algorithm ($MD$)

```
1: Call DCR(Pk)
2: for (j = 1 to n_pcs) do
3:    Calculate CT_1^j
4:    Calculate CT_2^j
5:    if (CT_1^j < CT_2^j) then
6:       R_1 is selected
7:    else
8:       if (CT_1^j > CT_2^j) then
9:          R_2 is selected
10:      else
11:         if (CT_1^j > CT_2^j) then
12:            Calculate Ii1
13:            Calculate Ii2
14:            if (Ii1 ≤ Ii2) then
15:               R_1 is selected
16:            else
17:               R_2 is selected
18:            end if
19:         end if
20:      end if
21:   end if
22: end for
23: Calculate Tt_max
24: Return Tt_max
```

## Random-search and insertion time algorithm ($\widetilde{SI}$)

The idle interval is better to be avoided because these intervals can give a bad result. Thus, in this algorithm, some packets are inserted in these idle intervals when the constraints allow the scheduling. Firstly, the schedule of ($DT$) is given. The sets of idle intervals are denoted as $I1$ and $I2$ on the first and second router, respectively. The idle intervals in the first and second routers are $n_1$ and $n_2$. The non-scheduled packet is inserted in the idle time $I1$ by looping until $n_1$. If there is no possibility of scheduling this packet, the non-scheduled packet is inserted in the second router in the idle time $I2$ by looping until $n_2$. After obtaining the final schedule, the received sequence will be used as the initial solution for applying ($ER2$). This algorithm denotes $\widetilde{SI}$. Hereafter, the procedure that sorts a list $L$ given as input relies on the transmission time-decreasing order denoted as ICR($L$). The procedure that searches and fixes the idle intervals denotes SF(). The starting time of the idle interval is determined by the procedure SIT(). Feas($R$) symbolizes the function that can detect the feasibility of scheduling the selected packet on the router $R$. This function returns "True" if the schedule is feasible and "False" otherwise.

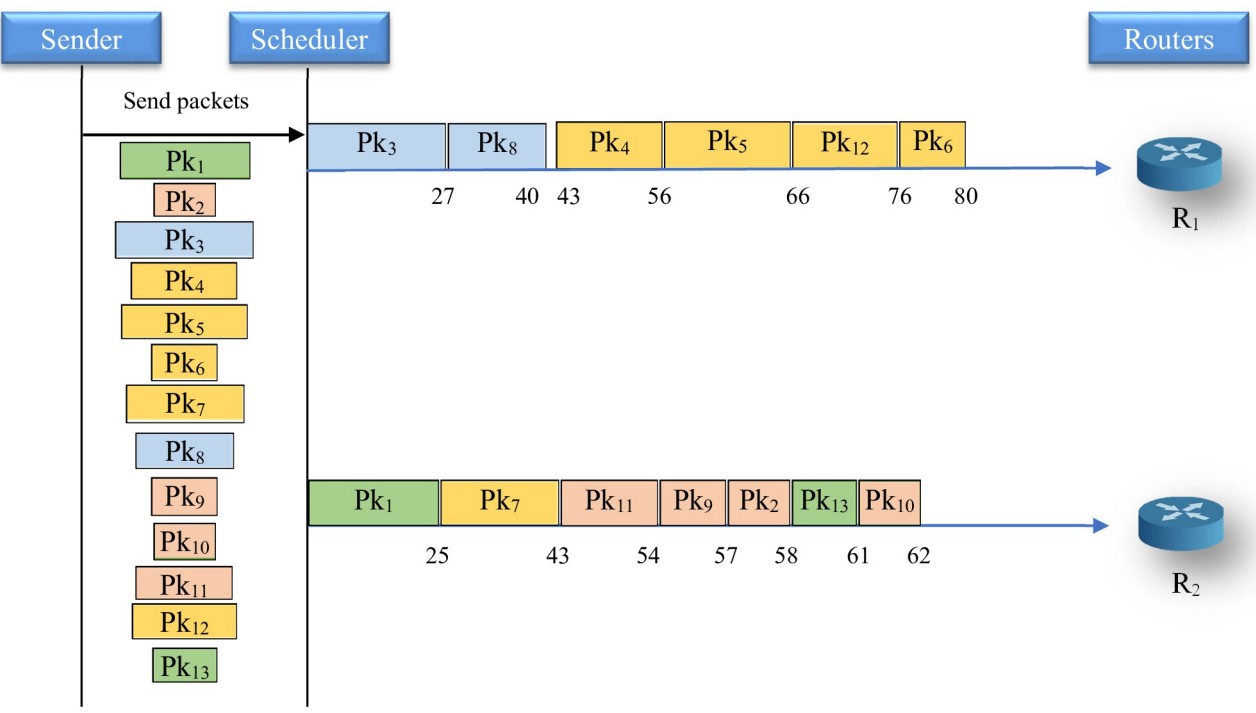

**Fig 2.**

**Algorithm 2** Search and Insertion Time Algorithm (SI)

```
1: Call ICR(PK)
2: Set n₁ = 0 and n₂ = 0
3: for (j = 1 to n_pcs) do
4:    Set check1 = 0 and check2 = 0
5:    Call SF()
6:    for (k = 1 to n₁) do
7:       Call SIT()
8:       if (Feas(1) = True then
9:          Set check1 + +
10:         Calculate CT₁ʲ
11:      end if
12:   end for
13:   for (k = 1 to n₂) do
14:      Call SIT()
15:      if (Feas(2) = True then
16:         Set check2 + +
17:         Calculate CT₂ʲ
18:      end if
19:   end for
20:   if (check1 ≠ 0 OR check2 ≠ 0) then
21:      Calculate CTʲ = min(CT₁ʲ, CT₂ʲ)
22:   end if
23: end for
24: Calculate Tt_max
25: Return Tt_max
```

## Random-critical security level algorithm ($\widetilde{CS}$)

All packets with the same security level are grouped in the set $PsL_i$ with $i = \{1, \ldots, n_{Sl}\}$. The sum of all $Pt_j$, $\forall Pt_j \in PsL_i$ is denoted by $SSL_i$. The critical security level denoted by $CL$ is the security level with the maximum value of $SSL_i$, $\forall i = \{1, \ldots, n_{Sl}\}$. The fictive packet denoted by $PF_i$, $i = \{1, \ldots, n_{Sl}\}$, is a new packet with a transmission time $SSL_i$. The fictive packets are sorted based on their $SSL_{i-}$ decreasing order, $\forall i = \{1, \ldots, n_{Sl}\}$. In each $PsL_i$, the packets are sorted according to $DT$. Next, the total time $T$ is calculated as $\sum_{i=1}^{n_{Sl}} SSL_i$. This researched problem lower bound can be taken as $\frac{T}{2}$ and denoted by $LB$. Next, the sequence of packets $PsL_i$ will be used to schedule packets on the first router until reaching $LB$. After that, the leftover packets will be scheduled on the second router. The acquired schedule will construct a new sequence that will be utilized to apply the ($ER2$). Example illustrates the obtaining of the $CS$ sequence.

**Example 2** *The same instance detailed in* Table 1 *is considered. Firstly, the packets are sorted in each set $PsL_i$ with $i = \{1, \ldots, n_{Sl}\}$ according to their transmission time- order.* Table 3 *shows the generation of the four fictive packets.*

*The next step in applying the algorithm is sorting the fictive packets, as described above. Indeed, the sequence to be scheduled on the two routers is listed as follows $\{PsL_4, PsL_3, PsL_1, PsL_2\}$. This means that the sequence is $\{\{7,4,5,12,6\},\{3,8\},\{1,13\},\{11,9,2,10\}\}$. After that, the packets are scheduled in the last sequence until reaching LB on the first router. For this stage, the scheduled packets on $R_1$ are $\{7,4,5,12,6,8,2\}$ and $\{3,1,13,11,9,10\}$ on $R_2$. This schedule gives the values $Tt_1 = 69$ and $Tt_2 = 72$ with a total transmission time of $Tt_{max} = 72$. The CS sequence is $\{7,4,5,12,6,8,2,3,1,13,11,9,10\}$. This sequence is the initial solution for ($ER2$).*

## Random packet-classification first variant algorithm ($\widetilde{PF}$)

The first step is to divide the packets into three groups $G_1, G_2$, and $G_3$. The first $\frac{n_{pcs}}{2}$ packets will be assigned to $G_1$. At the same time, the second $\frac{n_{pcs}}{2}$ packets will be assigned to $G_2$. The remaining packets will be assigned to $G_3$. The second step is sorting packets, in each group, according to their transmission time-increasing order. Finally, the sequence for scheduling packets is $G_2$, $G_1$, and $G_3$. After obtaining the final schedule, the obtained sequence is the initial solution for applying ($ER2$). This algorithm denotes $\widetilde{PF}$.

## Random packet-classification second variant algorithm ($\widetilde{PS}$)

The first step is to divide the packets into three groups $G_1, G_2$, and $G_3$. The first $\frac{n_{pcs}}{2}$ packets will be assigned to group $G_1$. At the same time, the second $\frac{n_{pcs}}{2}$ packets will be assigned to group $G_2$. The remaining packets will be assigned to $G_3$. The second step is sorting the packets in each group following their transmission time-increasing order. Finally, the sequence to schedule packets is $G_1$, $G_3$, and $G_2$. After obtaining the final schedule, the obtained sequence is the initial solution for applying ($ER2$). This algorithm denotes by $\widetilde{PS}$.

**Table 3. Generation of the fictive packets.**

|  | $i$ | $j$ |  |  |  |  | $SSL_i$ |
|---|---|---|---|---|---|---|---|
| $PsL_i$ | 1 | 1 | 13 |  |  |  | 28 |
|  | 2 | 11 | 9 | 2 | 10 |  | 15 |
|  | 3 | 3 | 8 |  |  |  | 40 |
|  | 4 | 7 | 4 | 5 | 12 | 6 | 55 |

### Best-random algorithm ($\widetilde{PS}$)

Firstly, the algorithms $\widetilde{CS}$ and $\widetilde{SI}$ are called. The solutions returned by $\widetilde{CS}$ and $\widetilde{SI}$ denote $T_{CS}$ and $T_{SI}$, respectively. The value returned by $\widetilde{BR}$ is $T_{BR} = min(T_{CS}, T_{SI})$. The algorithm $\widetilde{CS}$ and $\widetilde{SI}$ are designed in the functions denoted by $PrCSt()$ and $PrSIt()$, respectively. This algorithm of $\widetilde{BR}$ is described below. The random-critical security level algorithm ameliorates the scheduling based on the use of the critical security level. The random search and insertion time algorithm contribute to ameliorating the results by using the insertion method that reaches a better result. The $\widetilde{BR}$ is constructed by the maximum of these latter algorithms. The choice is based on the non-dominance of the proposed algorithms. In fact, $\widetilde{CS}$ and $\widetilde{SI}$ are non-dominants.

## Results and discussion

The section presents the setup, experimentation, and results concerning the proposed algorithms. C++ is used to implement the proposed algorithms. Five classes are coded and presented to show the algorithms' performance. The experimental results were obtained using: (i) a personal computer with a microprocessor i5, (ii) 8 GB of RAM, and (iii) Windows 10 as an operating system. This paper uses two types of distribution uniform and binomial distributions. The distribution is used to generate the values of the transmission time of a packet. $UD[b, f]$ denotes uniform distribution and $BD[b, f]$ for the binomial distribution with $b$ as the minimum value for $Pt_j$ and $f$ as the maximum value for $Pt_j$. The five classes that generate the different values of the $Pt_j$ are detailed as follows:

- Class 1: C1: $UD[2, 10]$;

- Class 2: C2: $UD[6, 15]$;

- Class 3: C3: $UD[5, 25]$;

- Class 4: C4: $BD[1, 20]$;

- Class 5: C5: $BD[1, 30]$.

The number of packets is defined as {5, 12, 15, 20, 25, 40, 60, 120}. The security level numbers are defined in {2, 3, 4, 5, 6}. Hence, in total, we have $1 \times 2 \times 5 \times 10 + 7 \times 5 \times 5 \times 10 = 1850$ instances. The metrics used to perform the proposed algorithms are:

- $\hat{T}$ is the minimum value of $Tt_{max}$ for all algorithms;

- $T$ is the $Tt_{max}$ which is the developed algorithms returned value;

- $Pc$ is the instances percentage when $\hat{T} = T$;

- $Gp = \frac{T - \hat{T}}{\hat{T}}$ represents the gap between the developed algorithm and the best-obtained value;

- $AgP$ is the average of $Gp$ over a group of instances;

- $Time$ represents in seconds the average execution time. The symbol "–" marks when the execution time is less than 0.001 s.

This section compares the proposed algorithms using the metrics detailed above and the generated class of instances. First, a compression of the proposed algorithms is presented. Next, the results of the developed algorithms are discussed in comparison with the results of the algorithms developed in the literature [12–14]. An overview of the results for all proposed

**Table 4. Overview results of all proposed algorithms.**

|  | $\widetilde{DT}$ | $\widetilde{MD}$ | $\widetilde{SI}$ | $\widetilde{CS}$ | $\widetilde{PF}$ | $\widetilde{PS}$ | $\widetilde{BR}$ |
|---|---|---|---|---|---|---|---|
| Pr | 73.7% | 89.3% | 89.9% | 62.7% | 60.8% | 67.1% | 95.1% |
| Dv | 0.004 | 0.002 | 0.001 | 0.007 | 0.008 | 0.004 | 0.001 |
| Time | 0.000 | 0.000 | 0.000 | 0.028 | 1.172 | 0.001 | 0.028 |

algorithms is shown in Table 4. The result reveals that $\widetilde{BR}$ is the best with a 95.1% percentage, 0.001 as a gap, and 0.028 s as an average time. The second-best one is $\widetilde{SI}$, with a percentage of 89.9%.

The best algorithm proposed in the literature [12] is the Modified Decreasing Estimated-Transmission Time Algorithm (*MDETA*). While the best algorithm in [13] is the Randomized Longest Transmission time first algorithm ($\overline{RLT}$). On the other hand, the best algorithm proposed in [14] is the first variants of the Random-Grouped Classification with Shortest Scheduling Algorithms ($RGS_1$). Table 5 shows an overview of the performance results for the developed algorithms in the literature. A comparison was accomplished on each value of the three best-proposed algorithms in the literature namely *MDETA*, $\overline{RLT}$, and $RGS_1$. Results presented in Table 5 point out that the best algorithm in literature is *MDETA*, with a percentage of 82%, compared to $\overline{RLT}$, with a percentage of 48%, and $RGS_1$ with a percentage of 28%. In Table 5, the *Dv* is calculated based on the best value of only the results of *MDETA*, $\overline{RLT}$, and $RGS_1$.

Based on the literature, the best algorithm is *MDETA* which will be compared to the best-proposed algorithm $\widetilde{BR}$. The best-returned value after running *MDETA* and $\widetilde{BR}$ is recorded in Table 6 and compared to each value. This latter table shows that $\widetilde{BR}$ is the best in 100% of cases. This latter table shows that the average gap of less than 0.001 is reached $\widetilde{BR}$. In Table 6, the *Dv* is calculated based on the best value of only the results of *MDET* and $\widetilde{BR}$.

Table 7 displays the *AgP* rates in all proposed algorithms during the changes of the $n_{pcs}$. It highlights that $\widetilde{BR}$ is the best one while reaching an average gap of less than 0.001 at four times as follows: $n_{pcs}$ = {5, 40, 60, 120}. While $\widetilde{DT}$, $\widetilde{CS}$, $\widetilde{PF}$, and $\widetilde{PS}$ reach an average gap of less than 0.001 at only one time where $n_{pcs}$ = 5. $\widetilde{MD}$ and $\widetilde{SI}$ reach an average gap of less than 0.001 when $n_{pcs}$ = {5, 120}. The maximum *AgP* value of 0.016 is reached for the algorithm $\widetilde{CS}$ when $n_{pcs}$ = 12.

**Table 5. An overview of the result from the literature on the developed algorithms.**

| 2-4 | *MDETA* | $\overline{RLT}$ | $RGS_1$ |
|---|---|---|---|
| Pr | 82% | 48% | 28% |
| Dv | 0.007 | 0.015 | 0.037 |
| Time | - | - | 0.001 |

**Table 6. An overview of the result of the best-proposed algorithm and the best from the literature.**

| 2-3 | *MDETA* | $\widetilde{BR}$ |
|---|---|---|
| Pr | 48% | 100 |
| Dv | 0.015 | 0 |
| Time | - | 0.028 |

Table 7. The *AgP* values where $n_{pcs}$ change for all algorithms.

| $n_{pcs}$ | $\widetilde{DT}$ | $\widetilde{MD}$ | $\widetilde{SI}$ | $\widetilde{CS}$ | $\widetilde{PF}$ | $\widetilde{PS}$ | $\widetilde{BR}$ |
|---|---|---|---|---|---|---|---|
| 5 | 0.000 | 0.000 | 0.000 | 0.000 | 0.000 | 0.000 | 0.000 |
| 12 | 0.003 | 0.002 | 0.001 | 0.016 | 0.010 | 0.004 | 0.001 |
| 15 | 0.007 | 0.006 | 0.003 | 0.011 | 0.013 | 0.007 | 0.002 |
| 20 | 0.002 | 0.002 | 0.001 | 0.008 | 0.009 | 0.004 | 0.001 |
| 25 | 0.005 | 0.004 | 0.002 | 0.006 | 0.009 | 0.003 | 0.001 |
| 40 | 0.003 | 0.002 | 0.001 | 0.004 | 0.004 | 0.005 | 0.000 |
| 60 | 0.004 | 0.002 | 0.001 | 0.003 | 0.006 | 0.004 | 0.000 |
| 120 | 0.004 | 0.000 | 0.000 | 0.002 | 0.005 | 0.005 | 0.000 |

The variation of the average running time according to the number of packets for $\widetilde{BR}$ is illustrated in Fig 3 demonstrates the increases in time whenever there is an increase in the number of packets. In addition, the maximum time value is 0.066 s reached when $n_{pcs}$ = 120.

Table 8 presents the *AgP* values where $n_{Sl}$ changes for all algorithms. This table shows that the minimum average gap of less than 0.001 is reached by $\widetilde{BR}$ when $n_{Sl}$ = 2, for $\widetilde{SI}$ and $\widetilde{MD}$ when $n_{Sl}$ = 6. The maximum average gap of 0.014 is reached by $\widetilde{PF}$ when $n_{Sl}$ = 2.

The variation of the average running time according to the number of security levels for $\widetilde{BR}$ is illustrated in Fig 4 demonstrates the increases in time whenever there is an increase in the number of security levels. In addition, the maximum time value is 0.033 s reached when $n_{Sl}$ = 6.

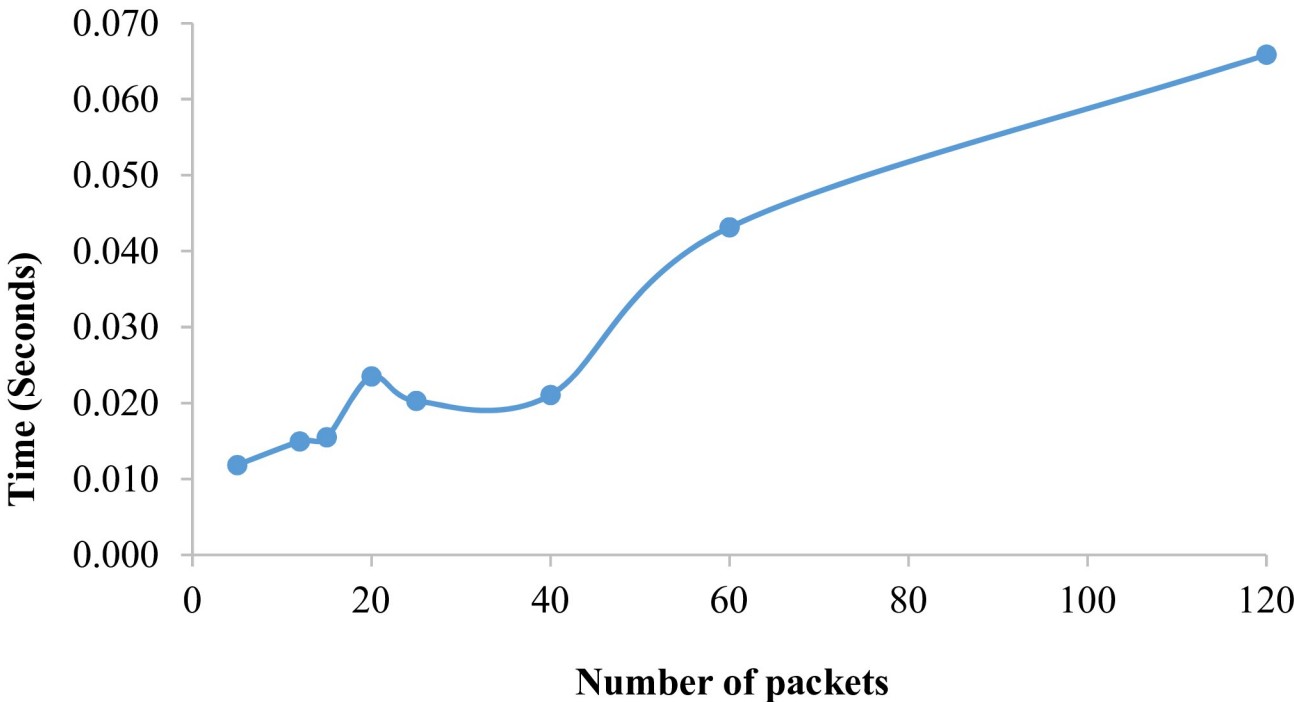

Fig 3.

**Table 8. The *AgP* values where $n_{SI}$ changes for all algorithms.**

| $n_{SI}$ | $\widetilde{DT}$ | $\widetilde{MD}$ | $\widetilde{SI}$ | $\widetilde{CS}$ | $\widetilde{PF}$ | $\widetilde{PS}$ | $\widetilde{BR}$ |
|---|---|---|---|---|---|---|---|
| 2 | 0.008 | 0.005 | 0.002 | 0.009 | 0.014 | 0.008 | 0.001 |
| 3 | 0.004 | 0.002 | 0.001 | 0.007 | 0.009 | 0.005 | 0.001 |
| 4 | 0.003 | 0.002 | 0.001 | 0.009 | 0.008 | 0.004 | 0.001 |
| 5 | 0.002 | 0.002 | 0.001 | 0.007 | 0.006 | 0.003 | 0.001 |
| 6 | 0.001 | 0.000 | 0.000 | 0.001 | 0.001 | 0.001 | 0.000 |

## Conclusion

This research investigates the problem of transmitting multilevel secure data based on a security constraint through routers such that packets belonging to the same security levels cannot, in any case, be transmitted through the two routers simultaneously. This problem is an NP-hard problem. Seven algorithms are proposed to resolve the presented problem. The performance measurements of the proposed algorithms show that the Best-Random Algorithm ($\widetilde{BR}$) is the most efficient. Furthermore, comparing $\widetilde{BR}$ with the previous best result presented by *MDETA* shows that $\widetilde{BR}$ is the best, with 0.028 s and an average gap of less than 0.001. The future directions of this research go in three ways. The first one is utilizing the proposed routine on other NP-hard problems. The second way is the development of a lower bound of the proposed problem that can give a better result for the proposed algorithms—finally, comparing the results obtained by the proposed routine with the one obtained by applying different metaheuristics like genetic algorithm and particle swarm optimization.

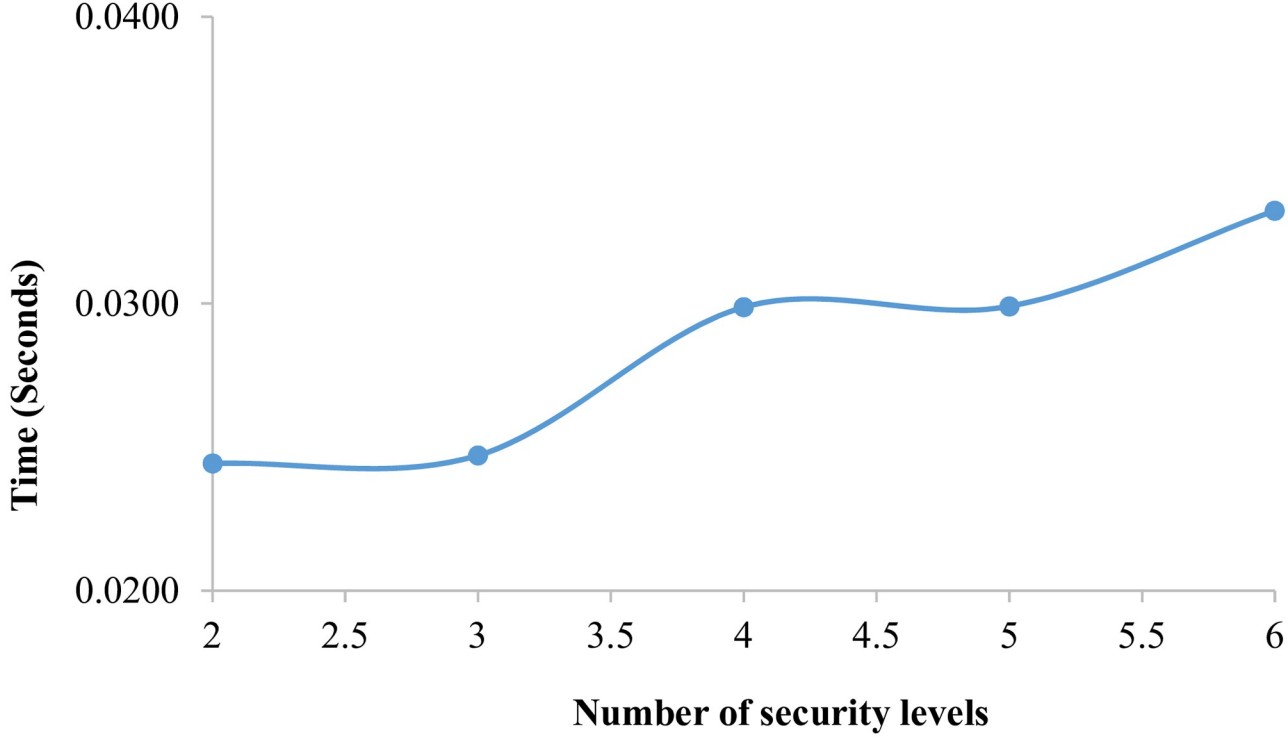

**Fig 4.**

## Author Contributions

**Conceptualization:** Mohammad Mahmood Otoom, Imen Achour, Mohamed Nazih Omri.

**Data curation:** Akram Y. Sarhan, Imen Achour, Mohamed Nazih Omri.

**Formal analysis:** Mohammad Mahmood Otoom, Akram Y. Sarhan, Mohamed Nazih Omri.

**Funding acquisition:** Mohammad Mahmood Otoom.

**Investigation:** Mohammad Mahmood Otoom, Akram Y. Sarhan.

**Methodology:** Mahdi Jemmali, Mohamed Nazih Omri.

**Project administration:** Mahdi Jemmali.

**Resources:** Mahdi Jemmali, Ibrahim Alsaduni.

**Software:** Mahdi Jemmali.

**Supervision:** Mahdi Jemmali.

**Validation:** Mahdi Jemmali, Imen Achour.

**Visualization:** Mohammad Mahmood Otoom, Mahdi Jemmali, Imen Achour, Ibrahim Alsaduni, Mohamed Nazih Omri.

**Writing – original draft:** Mohammad Mahmood Otoom, Mahdi Jemmali, Imen Achour, Ibrahim Alsaduni, Mohamed Nazih Omri.

**Writing – review & editing:** Mahdi Jemmali, Ibrahim Alsaduni, Mohamed Nazih Omri.

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
