## [Decision Letter · Decision Letter 0]

2 May 2023

PONE-D-23-10551Routine-Based Algorithms for a Secure Data Dissemination Based on Security-Category ConstraintsPLOS ONE

Dear Dr. Jemmali,

Thank you for submitting your manuscript to PLOS ONE. After careful consideration, we feel that it has merit but does not fully meet PLOS ONE’s publication criteria as it currently stands. Therefore, we invite you to submit a revised version of the manuscript that addresses the points raised during the review process.

 Please submit your revised manuscript by Jun 16 2023 11:59PM. If you will need more time than this to complete your revisions, please reply to this message or contact the journal office at plosone@plos.org. Please include the following items when submitting your revised manuscript:A rebuttal letter that responds to each point raised by the academic editor and reviewer(s). You should upload this letter as a separate file labeled 'Response to Reviewers'.A marked-up copy of your manuscript that highlights changes made to the original version. You should upload this as a separate file labeled 'Revised Manuscript with Track Changes'.An unmarked version of your revised paper without tracked changes. You should upload this as a separate file labeled 'Manuscript'.

We look forward to receiving your revised manuscript.

Kind regards,

Olugbemiga Solomon POPOỌLA, Ph.D.

Academic Editor

PLOS ONE

Journal Requirements:

4. We note you have included a table to which you do not refer in the text of your manuscript. Please ensure that you refer to Table 7 in your text; if accepted, production will need this reference to link the reader to the Table.

Additional Editor Comments:

Although the research idea is good, the ideas could NOT be sufficiently academically substantiated; because, motivations are largely NOT based on articulated Gaps in literature; datasets are fictive; and NO scientific experiment could be linked with the authors claims and the professed Results.  Thus, the authors could NOT satisfactorily ensure a CONSISTENT, COHERENT, and LOGICALLY organized ideas, which are scholarly presented, and scientifically proven with replicable methods.  Consequently, the decision to accept the manuscript (as it is) was put on hold in the best interest of the authors; as the issues raised are major, and may impact negatively on the Journal as well.  Therefore, authors should take diligent attention to the followings:In addition to the Reviewers' comments, the ATTACHED file is for the attention of the authors.  It contains addtional COMMENTS on the assessment of the manuscript.In the attachment, the issues raised are orderly organized under the respective Sections of a typical scholarly article.Apart from the issues raised, HINTS are provided for each Section; authors may consider any of them helpful in the course of effecting the revisions.For the sake of emphasis, few issues are repeated across the Sections; nonetheless, every issue should be addressed only once.Authors should NOTE that it is NOT enough to address issues, but that every issue addressed should be appropriately placed in the context of scientific writing.Therefore, comments should be addressed section-by-section; point-to-point; and responses MUST appear in RED PRINT (on the correction copy).  To ensure intra and inter Sections Consistencies and Coherencies, authors may rework the study, reorganize the presentation, and be academic in approach.Authors should be aware of the fact that this is an opportunity for revisions; and there may NOT be any other chance for such privilege.To avoid Rejection, please, a Point-to-Point responses, to the issues raised by the Reviewers (and the comments in the ATTACHED file), are REQUIRED.  As usual, both the corrected copy (Revised version 1), and the correction copy (Original version, with corrections physically marked out, in RED PRINT, on the manuscript), should be submitted for FINAL re-evaluations.

Reviewers' comments:

Reviewer's Responses to Questions

**Comments to the Author**

1. Is the manuscript technically sound, and do the data support the conclusions?

Reviewer #1: Partly

Reviewer #2: Yes

2. Has the statistical analysis been performed appropriately and rigorously? 

Reviewer #1: N/A

Reviewer #2: Yes

3. Have the authors made all data underlying the findings in their manuscript fully available?

Reviewer #1: Yes

Reviewer #2: Yes

4. Is the manuscript presented in an intelligible fashion and written in standard English?

Reviewer #1: No

Reviewer #2: Yes

5. Review Comments to the Author

Reviewer #1: 1. The abstract is too long, need to be summarised in such way to point out the main important factors and achievements in this research work.

2. In this “In the same context, the author in [16] proposed novel heuristics to solve the studied problem approximately” what is the solution that has been proposed?

3. In “In [19], the authors treated the cloud computing environments attacks.” What exactly means of attach and which kind?

4. In “However, there only three works dealt with the studied problem [14] [15] [16]. The studied problem is an NP-hard one. Working on such problem is a real challenge” the English need to improve and mentioned what kind of problem of NP-hard has it?

5. Figure 2 explanation is missing, I suggest the authors to explain more about graphs and figures as well as tables to make it more clear to readers.

6. The English need improvement for entire paper.

7. The conclusion seems need to be summarised as well to pint out the main achievements.

Reviewer #2: In the manuscript entitled “Routine-Based Algorithms for a Secure Data Dissemination Based on Security-Category Constraints” the problem of the transmission of multilevel secure data based on a security constraint through routers was investigated. The article is adequate for the journal but extensive revision is essential to make it acceptable for publication. The specific comments are appended below

• Carefully check and fix all the grammar mistakes starting from the first paragraph of the introduction.

• Many places in the article require referencing. For example:

(i) “The number of devices connected to the Internet significantly increases; for instance, more than ten billion devices are connected today”.

(ii) “In particular, IoT, digitalization, big data, and artificial intelligence are considered the primary causes of cybersecurity issues since they have notably increased internet connection use. As a result, attacks have rapidly expanded to target significant businesses and industries, including health organizations, to disrupt infrastructure and steal sensitive data”

These are not common knowledge that should not be backed up with reference(s). There are several similar places in the articles that referencing them is paramount. Correction should be effected accordingly.

• In many places, multiple references are cited together. For instance, references [6] [7] [8] [9] [10] [11] [12] [13], and [34] [35] [36] [37] [38] [39]. Please check the entire manuscript thoroughly and eliminate ALL the lumped citations. This should be done by characterizing the references individually. That is, mention 1 or 2 phrases per reference to show the individual contribution or how it is different from the others and why it deserves mentioning. Or else, cite the most relevant reference only.

• It has not been made apparent how novel this research is. What obstacles, problems, or gaps in the body of knowledge made the current study necessary? What sets this study apart from other related studies? This should be stated in clear terms.

• Seven algorithms have been proposed in this study. Why? More so, theories related to the proposed algorithms were highlighted without a single citation. Please acknowledge the sources of your literature accordingly.

• Reproducibility and transparency are two of the essential traits of good modern research. This study cannot easily be reproduced in its current form by interesting researchers. It might be challenging to distinguish between the different sections of the work, particularly the methods, findings, and discussion sections. The usual article format of an introduction, methodology, results and discussion, and the conclusion is encouraged for the Authors to follow. In the meanwhile, the article can be broken up into distinct, numbered sections and sub-sections.

• Only one Figure is presented in the results and discussion section. The unit of the variable on the Y-axis (Time) of Fig. 3 needs to be specified in seconds, minutes, or hours. Figures provide efficient visual presentations of your qualitative or quantitative findings. Therefore, to strengthen the results and discussion sections, it is encouraged to add more attractive figures to the manuscript.

• The conclusion section is well written.

6. PLOS authors have the option to publish the peer review history of their article (what does this mean?). If published, this will include your full peer review and any attached files.

Reviewer #1: No

Reviewer #2: **Yes: **Usman Alhaji Dodo

---

## [Author Response · Author response to Decision Letter 0]

4 Aug 2023

Dear Prof. Editor

I greatly appreciate the time and effort you and the reviewers have dedicated to providing valuable feedback on my manuscript. I also thank the reviewers for their insightful comments on my paper. This letter responds to the points raised by the reviewers.

We believe that my paper is now suitable for publication in your journal.

Sincerely, 

Reviewer 1

We want to thank Reviewer 1, who contributed to the paper with his remarks that improved the quality of the article.

Reviewer's comments 1:

The abstract is too long, need to be summarised in such way to point out the main important factors and achievements in this research work.

Answer 1:

The abstract is totally rewritten as following:

“Sensitive data, such as financial, personal, or classified governmental information, must be protected throughout its cycle. This paper studies the problem of safeguarding transmitted data based on data categorization techniques. This research aims to use a novel routine as a new meta-heuristic to enhance a novel data categorization based-traffic classification technique where private data is classified into multiple confidential levels. As a result, two packets belonging to the same confidentiality level cannot be transmitted through two routers simultaneously, ensuring a high data protection level. Such a problem is determined by a non-deterministic polynomial-time hardness (NP-hard) problem; therefore, a scheduling algorithm is applied to minimize the total transmission time over the two considered routers. To measure the proposed scheme's performance, two types of distribution, uniform and binomial distributions used to generate packets transmission time datasets. The experimental result shows that the most efficient algorithm is the Best-Random Algorithm ($\\widetilde{BR}$), recording 0.028 s with an average gap of less than 0.001 in 95.1\\% of cases compared to all proposed algorithms. In addition, $\\widetilde{BR}$ is compared to the best-proposed algorithm in the literature which is the Modified decreasing Estimated-Transmission Time algorithm ($MDETA$). The results show that $\\widetilde{BR}$ is the best one in 100 \\% of cases where $MDETA$ reaches the best results in only 48 \\%.”

Reviewer's comments 2:

In this “In the same context, the author in [16] proposed novel heuristics to solve the studied problem approximately” what is the solution that has been proposed?

Answer 2:

The second paragraph in the subsection “Related Works” will be:

“In the same context, the author in [16] proposed novel heuristics to solve the problem of secure multilevel data transmission. The author used several recognized and unknown algorithmic methods like iterative, randomization, and probabilistic methods. The achieved results that both proposed algorithms called RGS1 and RGS2 were better compared with previous proposed algorithms that deal with same studied problem. “

Reviewer's comments 3:

In “In [19], the authors treated the cloud computing environments attacks.” What exactly means of attach and which kind?

Answer 3:

Corrected to : “Finally, the authors in [19] addressed attacks affecting SDN and cloud computing environments.”

Reviewer's comments 4:

In “However, there only three works dealt with the studied problem [14] [15] [16]. The studied problem is an NP-hard one. Working on such problem is a real challenge” the English need to improve and mentioned what kind of problem of NP-hard has it?

Answer 4:

Corrected to:

“Several algorithms have been proposed to deal with network scheduling problems. However, algorithms presented in [14] [15] [16] dealt with the issue presented in this paper which is transmitting multiple levels of data based on a constraint. Such a problem is an NP-hard [16] because the minimization of total time for transmitting data through two routers in this paper is reduced from two parallel machines NP-Hard problem.”

Reviewer's comments 5:

Figure 2 explanation is missing, I suggest the authors to explain more about graphs and figures as well as tables to make it more clear to readers.

Answer 5:

The following paragrapg is added to explain Figure 2:

“The latter figure shows that on the first router, the packets {4,5,12,6} are scheduled, however on router 2 packet 7 is scheduled respecting the constraint of the data security level.”

Reviewer's comments 6:

The English need improvement for entire paper.

Answer 6:

Done. The entire paper is edited.

Reviewer's comments 7:

The conclusion seems need to be summarised as well to pint out the main achievements.

Answer 7:

The conclusion is rewritten as following:

This research investigates the problem of transmitting multilevel secure data based on a security constraint through routers such that packets belonging to the same security levels cannot, in any case, be transmitted through the two routers simultaneously. This problem is an NP-hard problem. Seven algorithms are proposed to resolve the presented problem. The performance measurements of the proposed algorithms show that the Best-Random Algorithm ($\\widetilde{BR}$) is the most efficient. Furthermore, comparing $\\widetilde{BR}$ with the previous best result presented by $MDETA$ shows that $\\widetilde{BR}$ is the best, with 0.028 s and an average gap of less than 0.001. The future directions of this research go in three ways. The first one is utilizing the proposed routine on other NP-hard problems. The second way is the development of a lower bound of the proposed problem that can give a better result for the proposed algorithms—finally, comparing the results obtained by the proposed routine with the one obtained by applying different metaheuristics like genetic algorithm and particle swarm optimization.

Reviewer 2

We want to thank Reviewer 1, who contributed to the paper with his remarks that improved the quality of the article.

In the manuscript entitled “Routine-Based Algorithms for a Secure Data Dissemination Based on Security-Category Constraints” the problem of the transmission of multilevel secure data based on a security constraint through routers was investigated. The article is adequate for the journal but extensive revision is essential to make it acceptable for publication. The specific comments are appended below

Reviewer's comments 1:

Carefully check and fix all the grammar mistakes starting from the first paragraph of the introduction.

Answer 1:

We proofread the manuscript and fixed many grammatical issues. 

Reviewer's comments 2

Many places in the article require referencing. For example:

(i) “The number of devices connected to the Internet significantly increases; for instance, more than ten billion devices are connected today”. [AA][bb]

(ii) “In particular, IoT, digitalization, big data, and artificial intelligence are considered the primary causes of cybersecurity issues since they have notably increased internet connection use. As a result, attacks have rapidly expanded to target significant businesses and industries, including health organizations, to disrupt infrastructure and steal sensitive data” [cc]

(iii) These are not common knowledge that should not be backed up with reference(s). There are several similar places in the articles that referencing them is paramount. Correction should be effected accordingly.

Answer 2:

We added the following required references. 

[AA] Rose K, Eldridge S, & Chapin L. 2015. The internet of things: An overview. The internet society (ISOC), 80, 1-50.

[BB] Sembroiz, D., Ricciardi, S., & Careglio, D. (2018). A novel cloud-based IoT architecture for smart building automation. In Security and Resilience in Intelligent data-Centric Systems and communication networks (pp. 215-233). Academic Press.

[cc] Jagatheesaperumal S K, Rahouti M, Ahmad K, Al-Fuqaha A & Guizani M. 2021. The duo of artificial intelligence and big data for industry 4.0: Applications, techniques, challenges, and future research directions. IEEE Internet of Things Journal, 9(15), 12861-12885.

Reviewer's comments 3:

In many places, multiple references are cited together. For instance, references [6] [7] [8] [9] [10] [11] [12] [13], and [34] [35] [36] [37] [38] [39]. Please check the entire manuscript thoroughly and eliminate ALL the lumped citations. This should be done by characterizing the references individually. That is, mention 1 or 2 phrases per reference to show the individual contribution or how it is different from the others and why it deserves mentioning. Or else, cite the most relevant reference only.

Answer 3:

Corrected.

Reviewer's comments 4:

It has not been made apparent how novel this research is. What obstacles, problems, or gaps in the body of knowledge made the current study necessary? What sets this study apart from other related studies? This should be stated in clear terms.

Answer 4:

Before the last paragraph in the inntorduction, we add the following paragraph:

“The novelty of this research is its security by design paradigm that initially relies on two routers for future network security. For example, the solution supports journalists for multilevel secure and successful transmission of their data in the form of network packets. Furthermore, the proposed work introduces several algorithmic solutions to deal with an NP-Hard problem in acceptable efficient time and use it in the network security field. Furthermore, compared with previous works. The proposed approach continues to enhance the results compared with previous developed algorithms \\cite{sarhan2021two} \\cite{sarhan2023novel} \\cite{sarhan2023novel2} results as it being detailed in the result and discussion section. It uses new algorithmic techniques and procedures. “

Reviewer's comments 5:

Seven algorithms have been proposed in this study. Why? More so, theories related to the proposed algorithms were highlighted without a single citation. Please acknowledge the sources of your literature accordingly.

Answer 5:

The following refernces are cited and the related results are compared with the proposed algorithms:

Jemmali M, Ben Hmida A. Quick dispatching-rules-based solution for the two parallel machines problem under mold constraints. Flexible Services and Manufacturing Journal. 2023; p. 1–26.

Sarhan A, Jemmali M, Ben Hmida A. Two routers network architecture and scheduling algorithms under packet category classification constraint. In: The 5th International Conference on Future Networks & Distributed Systems; 2021. p.119–127.

Sarhan A, Jemmali M. Novel intelligent architecture and approximate solution for future networks. Plos one. 2023;18(3):e0278183.

Reviewer's comments 6:

Reproducibility and transparency are two of the essential traits of good modern research. This study cannot easily be reproduced in its current form by interesting researchers. It might be challenging to distinguish between the different sections of the work, particularly the methods, findings, and discussion sections. The usual article format of an introduction, methodology, results and discussion, and the conclusion is encouraged for the Authors to follow. In the meanwhile, the article can be broken up into distinct, numbered sections and sub-sections.

Answer 6:

We modified the paper sections to match the article format.

Reviewer's comments 7:

Only one Figure is presented in the results and discussion section. The unit of the variable on the Y-axis (Time) of Fig. 3 needs to be specified in seconds, minutes, or hours. Figures provide efficient visual presentations of your qualitative or quantitative findings. Therefore, to strengthen the results and discussion sections, it is encouraged to add more attractive figures to the manuscript.

Answer 7:

The unit is added. Figure 4 is added.

Reviewer's comments 8:

The conclusion section is well written.

Answer 7:

Thank you.

Comments in the ATTACHED file

 SecureDissemination (PONE-D-23-10551) Reviews Report +1 

OBSERVED AIMS OF THE STUDY 

The authors seek to employ scheduling algorithms, with security category constraints, for data dissemination security. 

TOPICAL COMMENTS WITH SUGGESTIONS 

Title: Routine-Based Algorithms for a Secure Data Dissemination Based on Security-Category Constraints. 

Reviewer's comments 1:

1. Title is NOT clearly stated (somehow clumsy). 

2. SUGGESTION: The concepts upon which the security principle is based could be merged. Authors should NOTE that an unambiguous Title may attract better readership, for higher citations.

HINTS: modify the Title: in accordance with the study; prominently highlighting the GAP identified in the reviewed existing similar and/or related works. 

Answer 1:

The title is changed to :

“An Enhanced Multilevel Secure Data Dissemination Approximate Solution for Future Networks”.

Reviewer's comments 2:

Abstract: 

The Abstract can NOT stand alone; as an accurate summary of the research; because: 

1. NO specific weaknesses of the existing related works are mentioned. 

2. The “different datasets” are NOT stated. 

3. The algorithms used are NOT specified. 

4. Which algorithms are the “other proposed algorithms”? 

5. Which algorithm is “the best algorithm in the literature”? 

6. Are the objects of comparisons outcomes/outputs of similar basic experimental conditions/environments? 

7. The concluding statement does NOT justify the potential preferences of the proposed solution over the motivating limitations of the existing solutions. 

HINTS: restructure the Abstract into: why we must do what, what was actually done, how it was done, what result was obtained, and why the result was better; and present in line with the Journal format. 

Answer 2:

The abstract is totally rewritten.

Reviewer's comments 3:

Keywords: 

1. In this paper, NO Keyword is stated. 

2. It is academic to present a list of Keywords immediately after the Abstract. 

HINTS: ensure the keywords reflect – the implicit and explicit intents of the Title of the article. 

Answer 3:

Keywords are inserted in the submission system.

Reviewer's comments 4:

Introduction: 

1. Generally, the Introduction is quite elaborate. However, the organization is NOT logical enough. 

Answer 4:

Introduction reorganized. 

Reviewer's comments 5:

For instance, the statement of the problem is expected to be the opening paragraph, because that is the rationale, which necessitated the proposed methods. 

2. WITHOUT citation, statements like “… ten billion devices are connected today” are considered BOGUS. 

Answer 5:

References are added:

@article{rose2015internet,

title={The internet of things: An overview},

author={Rose, Karen and Eldridge, Scott and Chapin, Lyman},

journal={The internet society (ISOC)},

volume={80},

pages={1--50},

year={2015},

publisher={Reston, VA}

}

@incollection{sembroiz2018novel,

title={A novel cloud-based IoT architecture for smart building automation},

author={Sembroiz, David and Ricciardi, Sergio and Careglio, Davide},

booktitle={Security and resilience in intelligent data-centric systems and communication networks},

pages={215--233},

year={2018},

publisher={Elsevier}

}

Reviewer's comments 6:

3. Industry 5.0, OR 5.0 Industry, OR 5.0 Industry Revolution, OR 5IR; be accurately CONSISTENT. 

Answer 6:

Corrected.

Reviewer's comments 7:

 4. NO specific security LEAKAGE is presented for NONE of the many data communication network issues raised in the single-paragraphed Introduction. a. NO specific Gap in literature is presented. What are their weaknesses/LIMITATIONS? What are scientifically responsible for these GAPs? 

 b. NO specific existing solution (performance) is described as NOT satisfactory. Why are the proposed approaches considered capable of addressing the identified Gaps (in the existing related works)? (provide scientific JUSTIFICATIONS). 

 c. Why NO algorithm, except scheduling, could represent the best solution for secure data dissemination is NOT justified. 

Answer 7:

The introduction is rewritten. 

Reviewer's comments 8:

 5. Would the propose approach modifies the existing OSI model; as the authors are worry that “the OSI model’s current design less effective in dealing with numerous attacks”? How? 

Answer 8:

The propose approach doeas not modify the existing OSI model. The scheduler module with the novel added constraint can reduce the attacks into networks.

Reviewer's comments 8:

Would the proposed methods functionally enhance Border Gateway Protocol; since the authors expressed concerns that “BGP is vulnerable to attacks”? How? 

Answer 8:

No. The sentence is modified to: “However, it does not maintain security and is vulnerable to attacks \\cite{wang2022high}\\cite{abdallah2015survey}”.

Reviewer's comments 9:

7. The rationale for proposing “… an intelligent private network solution that uses a security policy to provide a multilevel data dissemination control based on security category constraints …” is NOT justified. 

a. What is the “security policy”? Why is the policy considered “intelligent” and “private”? 

 b. How many “multilevel controls”? Which particular “transmission controls” are multileveled; and how? 

 c. What are the “security categories”? Which specific “category constraints” are applicable? 

 d. How “the security category constraints” constitute an NP-hard problem is NOT clarified. 

Answer 9:

The introduction is rewrtitten.

The security category constraints” constitute an NP-hard problem is proved in literature in the following reference:

@inproceedings{sarhan2021two,

title={Two routers network architecture and scheduling algorithms under packet category classification constraint},

author={Sarhan, Akram and Jemmali, Mahdi and Ben Hmida, Abir},

booktitle={The 5th International Conference on Future Networks \\& Distributed Systems},

pages={119--127},

year={2021}

}

This refernec is cited in the paper.

Reviewer's comments 10:

 8. AMBIGUITIES: “While on one hand, the elaboration of the proposed solution relies on the scheduling algorithms presented in [6] [7] [8] [9] [10] [11] [12] [13], which are planned later to be utilized and adapted to the studied problem. On the other hand, applying the security category constraints is considered an NP-hard problem, which means that no algorithm will always efficiently produce the exact correct answer on all inputs. For that reason, the scheduling algorithm could represent the best choice to get the best solution …” 

 9. How are those existing algorithms [6-13] “… utilized and adapted to the studied problem …”? 

Answer 10:

Corrected.

Reviewer's comments 11:

 10. If “…scheduling algorithm could represent the best choice to get the best solution in the current work …”, why/how are “… the security category constraints …” functionally integrated into the scheduling framework? 

Answer 11:

The security category constraints functionally are integrated into the scheduling framework to solve the NP-hard problem which is already proved to be NP-hard one.

 11. Why is a “two-router network” proposed? 

The two-router network is studied in the literature in the following reference:

@inproceedings{sarhan2021two,

title={Two routers network architecture and scheduling algorithms under packet category classification constraint},

author={Sarhan, Akram and Jemmali, Mahdi and Ben Hmida, Abir},

booktitle={The 5th International Conference on Future Networks \\& Distributed Systems},

pages={119--127},

year={2021}

}

This refernec is cited in the paper.

Reviewer's comments 12:

 12. “The proposed two-router solution has many benefits: (i) introduces a security-by-design future network idea that relies on a dissemination-based packet classification; (ii) applies various algorithmic techniques such as dispatching rules, local insertion search, randomization method, and lifting procedure to enhance the security and performance of the transmitted packets in the computer network area; (iii) presents an idea that can be implemented as a private network application to assist individual privacy protection in critical environments and circumstances.” “The disadvantages of the proposed method are as follows: (i) the times’ complexity incurred from relying on a two-router solution that is a reduction from an NP-hard two-machine problem which demands using more techniques to solve it using algorithms with 𝑂(𝑛3) heuristically. (ii) Developing an exact solution requires using a lower bound in a branch-and-bound algorithm.” a. The motivation for the proposed methods is NOT clear; as benefit (ii) above is a set of different modules of the main approach, whose implementation are largely NOT reported; benefit (iii) is a mere future research projection; disadvantage (i) presents NO basis for the 𝑂(𝑛3) time complexity; and disadvantage (ii) supposed to be the target approach, since it is possible to give the exact solution. 

 b. NONE of the benefits NOR disadvantages is linked to any GAP in literature. 

HINTS: begin the Introduction with – a clear statement of the problem; how others have tried to solve the problem; how this paper improves on previous solutions. The motivation should be clearly highlighted, while showing the importance of the research. Strong and relevant key references are cited accordingly. Background scientific anatomical information is provided. The last statement should describe specifically what was done. 

Answer 12:

The introduction is rewritten. 

Reviewer's comments 13:

Literature Review (Related Works): 

1. The Literature reviews are generally vaguely presented; NOT concise; and, NOT coherent. 

2. N.B.: Reviews are expected to be presented either literature-by-literature, or concept-by-concept, or principle-by-principle, or theory-by-theory, or approach-by-approach (each paragraph containing a single unit of discussion). 

3. The methods for all the directly related works (that are relevant to the proposed methods) should be explicitly presented; especially, [14-16] supposed to be elaborately detailed; as the “works dealt with the studied problem” (according to the authors). 

4. Review appraisals (which should precisely elucidate the specific Gap that motivated the proposed approach) are NOT presented. Thus, the necessity-proof for the proposed solution is NOT scientifically justified. 

HINTS: organise each review into – motivations, objectives, methods, results, strengths (knowledge advanced), and weaknesses (limitations – GAPS) of the work. 

Methodology: 

1. How the approach minimizes transmission time is clear; as claimed in “… minimize the total transmission time over the two considered routers …”; but how such speed could translate to security is NOT presented. 

2. NONE of the datasets generated is presented; as being claimed in “… experiments were conducted with different datasets generated according to uniform and binomial distributions …” 

3. NO procedure is presented to justify that “… transmitted data is classified into different confidentiality levels … using … a non-classic (distinctive characteristics of data categories) traffic classification approach …” 

4. NONE of the statistical properties (for statistical classification) is scientifically justified as a security 

attribute; as being claimed in “… utilising the statistical properties of the network traffic flow … for security purposes such as filtering traffic and identifying and detecting malicious activity …” 

5. NO specific feature NOR features extraction procedure is presented; as being claimed in “… effectiveness of the classifier in statistical classification depends on the features extracted from the flow …” 

6. NO procedure is presented on how “… two packets belonging to the same confidentiality level can’t be transmitted through the two routers simultaneously …”; in furtherance to the authors claims. 

7. What parameters are the determinants for each security level, and the number of possible security levels? 

8. What determines the size of each packet? Are there limits to the possible number of packets? 

9. What is the security constraint that prevents simultaneous transmission of data? 

10. If “𝑃𝑐𝑠=𝑃𝑐𝑠1∪𝑃𝑐𝑠2”; then, “𝑛𝑝𝑐𝑠=|𝑃𝑐𝑠1|∪|𝑃𝑐𝑠2|” is NOT accurate (NOT a correct math formulation). 

11. Must the “security level” be always imposed and linked by the administrator? What are the standard guiding principles that the administrator must follow? 

12. Upon which principle does the “any heuristic ℎ()” operate? 

13. What are the parameters that stipulate the “probability 𝛽”? 

14. Until Example 1, there is NO mention of “the decreasing time algorithm”: INCOHERENCY. 

15. NONE of the seven proposed algorithms could appropriately characterise data transmission security; as DT (upon which all the remaining six algorithms are based) merely considers how long a process takes to complete: schedules from the longest process completion time to the shortest, engages the best possible number of processors, and maintains the order of the priority list, while possible idle times are minimized. 

16. How the 𝐸𝑅2 generation of sequences that are 𝐷𝑇−𝑏𝑎𝑠𝑒𝑑 translate to data protection of is NOT clear. 

17. Algorithm 1 should be revised to reflect the independence of 𝐼𝑖 on 𝐶𝑇1>𝐶𝑇2. 

 18. All examples should be within the context of proposed methods; since, according to the authors, “… experiments were conducted with different datasets generated …”. a. Procedures should be described using datasets that are relevant to each experiment (fictitious datasets would NOT work in this study). 

 b. All the examples, using fictive data, are evidences that NO experiment was carried out. 

Authors should ensure that: Methods answer ANY question around the aims and objectives of the study. Some of such questions are raised in the Introduction Section above. Methods should NOT be literature about concepts, but ACTIONS using applicable concepts; Except for references to the basis of the resulting experimental setups. 

Answer 13:

The Related Works is rewritten. 

Reviewer's comments 14:

Results and Discussions: 

1. Results that are outcomes of fictive data would definitely be fictitious. 

2. Thus, the results being claimed by the authors are NOT scientific, NOT academic, and NOT scholarly. 

Authors should ensure Results discussions MUST answer the following/related questions: What are the Findings? Which Finding justifies the expected outcomes? With what metrics were the quantity/quality measured? In the chosen subject DOMAIN, did the results meet the scope defined? Are there intermediate Results? Which background concepts are instrumental to the good outcome? Against which known work were the accuracy correlated? Is this outcome better? Does that outcome justify the motivating GAB? Whether the outcome is better or not, what future researches are suggested? (OPTIONAL). 

Discussions should be Relevant to – the Results and Findings in the context of the Topic’s objectives vis-à-vis related works; with answers to the research questions. 

Answer 14:

The results are not based on fictive data, but on data already proposed in the literature:

@inproceedings{sarhan2021two,

title={Two routers network architecture and scheduling algorithms under packet category classification constraint},

author={Sarhan, Akram and Jemmali, Mahdi and Ben Hmida, Abir},

booktitle={The 5th International Conference on Future Networks \\& Distributed Systems},

pages={119--127},

year={2021}

}

@article{sarhan2023novel,

title={Novel intelligent architecture and approximate solution for future networks},

author={Sarhan, Akram and Jemmali, Mahdi},

journal={Plos one},

volume={18},

number={3},

pages={e0278183},

year={2023},

publisher={Public Library of Science San Francisco, CA USA}

}

@article{sarhan2023novel2,

title={A novel smart multilevel security approach for secure data outsourcing in crisis},

author={Sarhan, Akram Y},

journal={PeerJ Computer Science},

volume={9},

pages={e1367},

year={2023},

publisher={PeerJ Inc.}

}

Reviewer's comments 15:

Conclusions: 

1. Emphasis on the Gaps in the previous related works, how the present study mitigates the Gaps, and the comparisons of their performances are NOT presented. 

2. The Findings of the Results, which purportedly support the conclusions, are NOT outcomes of replicable experimental setups. 

3. The suggested future researches are supposed to be the main target approaches of this study. 

Authors should give expositions on the Findings in the conclusion; by: reiterating the motivations and the identified gabs in a bid to declare the solutions being provisioned, while showcasing the additional knowledge contributions to the domain; giving justifications to whatsoever limitations identified, with suggestions for future works. 

Answer 15:

The conclusion is rewritten.

Reviewer's comments 16:

References/Citations: 

1. References are adequate; largely relevant; and just 68% recent. 

2. Citations are orderly, and appropriately formatted. 

Answer 16:

Corrected.

Reviewer's comments 17:

Authors should be mindful of: the Journal requirements; and simply strictly adhere to whatever citation and/or reference styles stipulated for submission and publishing processes. 

SPECIFIC COMMENTS AND SUGGESTED MODIFICATIONS 

1. Readership: is NOT good enough, as there are: 

 a. Language (grammar) concerns that needs attentions; 

 b. Indiscriminate use of “etc”; 

 c. Poor paragraphing (as in the Introduction and Related works sections); 

 d. Sections are being referred to by non-existing numbers, as sections are NOT numbered; 

 e. Some mnemonics are NOT defined at first instance (such as “PA”, “NP-hard”, “MDETA”). 

3. Manuscript Format: Please, adhere strictly to the Journal formats (e.g., title, authors/affiliations, abstract, fonts/font-sizes, alignments, paragraphing, keywords, section-heads, subheads, sub-subheads, citations, tables, lists/bulleting, abbreviations, figures, graphics, equations, reference listings). 

RECOMMENDATIONS 

The research idea is good. However, ideas could NOT be academically substantiated; motivations are largely NOT based on articulated Gaps in literature; datasets are fictive; and NO scientific experiment could be linked with the authors claims in the Introduction and the professed Results. Hence, the authors are encouraged to pay diligent attentions to the above comments, to ensure a CONSISTENT, COHERENT, and LOGICALLY organized ideas, which are scientifically proven and scholarly presented.

Answer 17:

The paper is revised, please see the tracking version.

---

## [Decision Letter · Decision Letter 1]

29 Aug 2023

PONE-D-23-10551R1An Enhanced Multilevel Secure Data Dissemination Approximate Solution for Future NetworksPLOS ONE

Dear Dr. Jemmali,

Thank you for submitting your manuscript to PLOS ONE. After careful consideration, we feel that it has merit but does not fully meet PLOS ONE’s publication criteria as it currently stands. Therefore, we invite you to submit a revised version of the manuscript that addresses the points raised during the review process.

The paper has undergone considerable revisions.  However, the points raised in the ATTACHED document file require more adequate attentions.  Additionally, as observed regarding the first [16] (before [1]) and the first [22] (after [20]) citations, ensure all apparent MISPLACEMENTS and duplications are appropriately addressed.

We look forward to receiving your revised manuscript.

Kind regards,

Olugbemiga Solomon POPOỌLA, Ph.D.

Academic Editor

PLOS ONE

Journal Requirements:

Reviewers' comments:

Reviewer's Responses to Questions

**Comments to the Author**

1. If the authors have adequately addressed your comments raised in a previous round of review and you feel that this manuscript is now acceptable for publication, you may indicate that here to bypass the “Comments to the Author” section, enter your conflict of interest statement in the “Confidential to Editor” section, and submit your "Accept" recommendation.

Reviewer #1: All comments have been addressed

Reviewer #2: All comments have been addressed

2. Is the manuscript technically sound, and do the data support the conclusions?

Reviewer #1: Yes

Reviewer #2: Yes

3. Has the statistical analysis been performed appropriately and rigorously? 

Reviewer #1: Yes

Reviewer #2: Yes

4. Have the authors made all data underlying the findings in their manuscript fully available?

Reviewer #1: Yes

Reviewer #2: Yes

5. Is the manuscript presented in an intelligible fashion and written in standard English?

Reviewer #1: Yes

Reviewer #2: Yes

6. Review Comments to the Author

Reviewer #1: Thanks for the corrections and responding all concerns.

most of the comments has been explained well and addressed accordingly.

Reviewer #2: The manuscript has been improved significantly to meet the publication standards. Meanwhile, the referencing of the supporting literature was done inaccurately. For instance, it is wrong to ascribe [16] to the first paper cited in the introduction. Therefore, authors are advised to adopt the reference style specified on the journal’s portal.

7. PLOS authors have the option to publish the peer review history of their article (what does this mean?). If published, this will include your full peer review and any attached files.

Reviewer #1: No

Reviewer #2: **Yes: **Usman Alhaji Dodo

---

## [Author Response · Author response to Decision Letter 1]

7 Dec 2023

Querry:

The manuscript has been improved significantly to meet the publication standards. Meanwhile, the referencing of the supporting literature was done inaccurately. For instance, it is wrong to ascribe [16] to the first paper cited in the introduction. Therefore, authors are advised to adopt the reference style specified on the journal’s portal.

Answer:

Done

---

## [Editor Report · Decision Letter 2]

14 Dec 2023

An Enhanced Multilevel Secure Data Dissemination Approximate Solution for Future Networks

PONE-D-23-10551R2

Dear Dr. Jemmali,

We’re pleased to inform you that your manuscript has been judged scientifically suitable for publication and will be formally accepted for publication once it meets all outstanding technical requirements.

Kind regards,

Olugbemiga Solomon POPOỌLA, Ph.D.

Academic Editor

PLOS ONE

Additional Editor Comments (optional):

However, the points raised in the ATTACHED document file require more adequate attentions.  Additionally, as observed regarding the first [22] (after [20] citation), ensure all apparent MISPLACEMENTS and/or duplications are appropriately addressed.
---

## [Editor Report · Acceptance letter]

28 Jan 2024

PONE-D-23-10551R2 

PLOS ONE

Dear Dr. Jemmali, 

I'm pleased to inform you that your manuscript has been deemed suitable for publication in PLOS ONE. Congratulations! Your manuscript is now being handed over to our production team.

Kind regards, 

on behalf of

Dr. Olugbemiga Solomon POPOỌLA 

Academic Editor

PLOS ONE